# Experimental and theoretical model for the origin of coiling of cellular protrusions around fibers

Raj Kumar Sadhu [1,8] ✉, Christian Hernandez-Padilla[2], Yael Eshed Eisenbach[3], Samo Penič[4], Lixia Zhang[5], Harshad D. Vishwasrao[5], Bahareh Behkam [2], Konstantinos Konstantopoulos[6], Hari Shroff[5,7], Aleš Iglič[4], Elior Peles [3], Amrinder S. Nain [2] ✉ & Nir S. Gov [1] ✉

Protrusions at the leading-edge of a cell play an important role in sensing the extracellular cues during cellular spreading and motility. Recent studies provided indications that these protrusions wrap (coil) around the extracellular fibers. However, the physics of this coiling process, and the mechanisms that drive it, are not well understood. We present a combined theoretical and experimental study of the coiling of cellular protrusions on fibers of different geometry. Our theoretical model describes membrane protrusions that are produced by curved membrane proteins that recruit the protrusive forces of actin polymerization, and identifies the role of bending and adhesion energies in orienting the leading-edges of the protrusions along the azimuthal (coiling) direction. Our model predicts that the cell's leading-edge coils on fibers with circular cross-section (above some critical radius), but the coiling ceases for flattened fibers of highly elliptical cross-section. These predictions are verified by 3D visualization and quantitation of coiling on suspended fibers using Dual-View light-sheet microscopy (diSPIM). Overall, we provide a theoretical framework, supported by experiments, which explains the physical origin of the coiling phenomenon.

Cellular protrusions play important roles in exploring and sensing the extracellular environment, during cell spreading and adhesion, cell migration, and cell–cell interaction[1–5]. Lamellipodia and filopodia are protrusive structures formed at the leading edge of a migratory cell[6–12]. These protrusions enable cells to adhere and spread on fiber-like surfaces[13–15], such as the fibers of the extracellular matrix (ECM)[16,17], as well as cylindrical protrusions of other cells, such as glial cells spreading over neighboring axonal extensions[18]. In vitro studies of the

cellular spreading and migration on fibers[13,14] have shown how different cell types organize on these fibers[19–24], with the cellular shape and motility found to depend on the curvature (diameter) of the fibers[15,19,21–26].

Experiments studying the membrane dynamics at the leading edge of cellular protrusions have found indications for coiling (wrapping) dynamics around extracellular fibers. In refs. 27,28, the leading edge of protrusions of metastatic cancer cells (breast and ovarian)

[1]Department of Chemical and Biological Physics, Weizmann Institute of Science, Rehovot 7610001, Israel. [2]Department of Mechanical Engineering, Virginia Tech, Blacksburg, VA 24061, USA. [3]Department of Molecular Cell Biology, Weizmann Institute of Science, Rehovot 7610001, Israel. [4]Laboratory of Physics, Faculty of Electrical Engineering, University of Ljubljana, Ljubljana, Slovenia. [5]Advanced Imaging and Microscopy Resource, National Institutes of Health, Bethesda, MD, USA. [6]Department of Chemical and Biomolecular Engineering, Johns Hopkins University, Baltimore, MD, USA. [7]Laboratory of High Resolution Optical Imaging, National Institute of Biomedical Imaging and Bioengineering, National Institutes of Health, Bethesda, MD, USA. [8]Present address: Institut Curie, PSL Research University, CNRS, UMR 168 Paris, France. ✉e-mail: raj-kumar.sadhu@curie.fr; nain@vt.edu; nir.gov@weizmann.ac.il

were observed to coil and rotate around the fiber's axis in a curvature-dependent manner, while in ref. 29 similar coiling dynamics of 'fin'-like protrusions were observed for several cell types (fibroblasts, epithelial, endothelial). We give below two examples of coiling dynamics of the leading edge of cellular protrusion: cells (Mouse Muscle Myoblasts, C2C12) extending protrusions along thin suspended artificial fibers (Fig. 1 and Supplementary Movies 1–3) and during the myelination process by glial cells on axons (Fig. 2 and Supplementary Movie 4). We discuss these experiments in detail below.

These examples emphasize the biological importance of the coiling process of cellular protrusions on fibers, such as for cell migration during tissue development, organogenesis, and cancer progression. The mechanisms that drives the tendency of the leading edge of cellular protrusions to rotate while spreading on fibers is not understood at present, and is the focus of this work.

Here we use a theoretical model[30] which describes a membrane vesicle spreading over an adhesive substrate, to address the question posed by the observations: why does the leading edge of cellular protrusions spread on fibers by coiling as opposed to simple axial extension? The basic concept underlying our model is that a population of membrane-bound protein complexes with convex curvature diffuse and aggregate on the membrane, while recruiting protrusive forces produced by actin polymerization. Examples of such protein complexes have been recently identified at the leading edge of cellular protrusions and ruffles, containing I-BAR proteins (such as IRSp53) and

nucleators of branched actin polymerization (such as the WAVE complex)[31,32]. Our model aims to expose the underlying physical principles that drive the membrane shape dynamics, using a minimal set of ingredients.

This model was shown to describe the spontaneous formation of membrane protrusions with lamellipodia-like leading edges. These leading edges form in our model due to spontaneous aggregation of the curved proteins that exert protrusive forces, which in the model represent the pressure exerted on the membrane by actin polymerization[30]. We show here that within this model, the leading edge of protrusions spontaneously reorient from axial to circumferential (coiling) orientation, when spreading on fibers. The model shows that minimization of adhesion and bending energies drives this reorientation and stabilizes the leading edge in the circumferential (coiling) direction. Overall, we provide a theoretical framework, supported by experiments, which suggests a very general physical origin for the coiling phenomenon.

Using Dual-View light-sheet microscopy (diSPIM)[33,34], we obtained high-resolution images of membrane ruffles that exhibit coiling dynamics at the leading edge of protrusions that cells extend along thin suspended artificial fibers (Fig. 1 and Supplementary Movies 1–3). The motion of the extended ruffles is easier to observe compared to the dynamics of the fiber-bound membrane protrusion, as indicated in Fig. 1C, D. The rotational directionality of both the coiling ruffles and the underlying membrane (kymograph in Fig. 1B) appear to coincide,

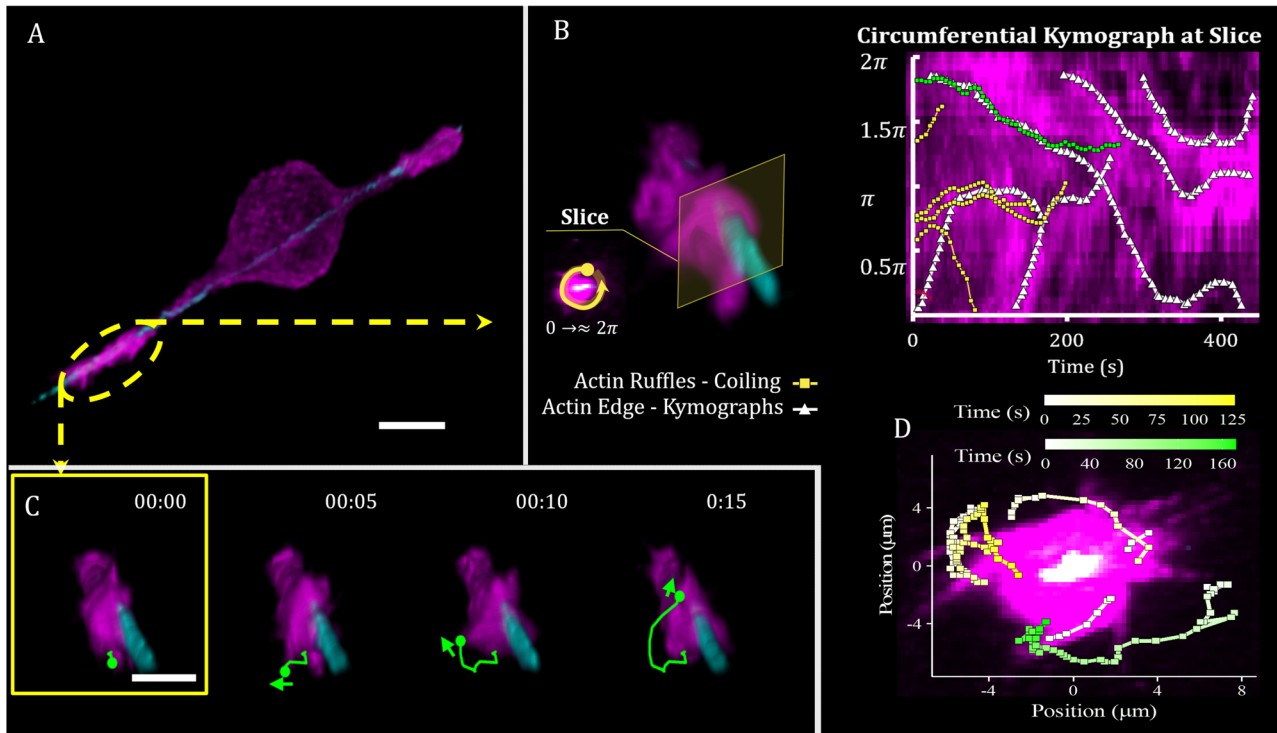

**Fig. 1 | Coiling events occur at the leading edges of cellular protrusions, extended by a cell that is suspended on a single fiber of 200-nm diameter. A** 3D render of a C2C12 cell captured with Dual-View Light Sheet microscopy. C2C12 cells were tagged with GFP actin (shown in magenta), and the polystyrene fiber was coated with Rhodamine Fibronectin (shown in cyan). The scale bar is 10 μm. **B** A zoomed-in section at the edge of the cellular protrusion, contained in the yellow oval frame in (**A**). A plane was selected from the fluorescent stack images corresponding to the indicated slice. Within this slice, a circular line was selected for a kymograph going from 0 to 2π, as shown in the left inset. Higher intensities in the kymographs (brighter magenta) represent regions richer in GFP actin as expressed by the cell. Tracing the regions of high actin intensity (white traces) in the circumferential kymograph illustrates the rotations of the cell membrane. The rotations can occur simultaneously in both the counter and clockwise directions. A

dominant rotation can be observed, making a nearly complete counterclockwise rotation (2π–0) taking ~350 s to complete. Shorter clockwise rotations (0–π and 0–1.4π) occur at 5 and 140 s, respectively. Coiling events shown in green and yellow traces represent the angular position of the events shown in (**D**), obtained by tracking the tips of the ruffles. The coiling event in green coincides with the counterclockwise rotation of the membrane ruffle in (**C**). **C** Strip of fluorescent images tracking of the region denoted by the yellow oval frame in (**A**). We trace the trajectory of a selected ruffle exhibiting coiling dynamics (green line), corresponding to the green trace in the kymograph in **B**. The scale bar shown in the first frame is 5 μm, and time stamp format is hh:mm. **D** Maximum intensity projection with the fiber oriented normal to the image plane (white area in the center), shown with overlayed data corresponding to the coiling events occurring at the selected edge of the cell. Source data are provided as a Source Data file.

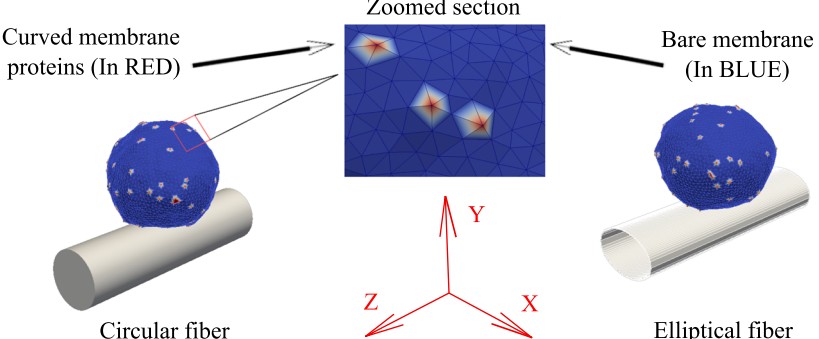

**Fig. 2 | Time-lapse imaging of mouse DRG myelinating culture showing initial contact between a thin process of pre-myelinating Schwann cell (green) spiraling around an axon (red).** Arrows point at clear spiral turns, as they rotate. Time intervals between images are 1 h. Images were subjected to 3D rendering using Zen 2012 (Carl Zeiss) to better emphasize the complex configuration. Scale bar is 20 μm, and time stamp format is hh:mm.

Zoomed section

Curved membrane proteins (In RED)

Bare membrane (In BLUE)

Y

Z        X

Circular fiber                                                          Elliptical fiber

**Fig. 3 | Schematic representation of our theoretical model.** We consider a three-dimensional vesicle, and an adhesive fiber of circular as well as elliptical cross-section. The red nodes on the vesicle represent the curved proteins, while the blue nodes are the bare membrane (which has no spontaneous curvature). A zoomed version of a small section of the vesicle surface is shown in the inset.

thus indicating the overall coiling of the cell membrane at the leading edge of the protrusions on the fibers. Note that the coiling dynamics of the ruffles is greatly diminished at the retracting, trailing edge of the cell. Therefore the coiling motion of the ruffles at the leading edge of the protrusions is closely correlated with the coiling motion of the leading edge as it spreads on the fiber[27]. Cells may form better overall grip when coiling on fibrillar structures (i.e., ECM and cellular processes), aiding cell motility and intercellular interactions. For example, cancer cells, which are highly motile, exhibit enhanced coiling activity on suspended fibers[27,28].

Another example for the coiling of cellular protrusions on fibers can be seen in the myelination process (Fig. 2 and Supplementary Movie 4). During the myelination process glial cells of the vertebrate central and peripheral nervous systems produce a multi-lamellar substance called myelin around axons, thereby allowing fast nerve conduction[35,36]. In the peripheral nervous system, Schwann cells myelination is a process that requires these cells to efficiently coil around fiber-like axons[37,38]. In order to closely visualize coiling of Schwann cells on axons, we used a transgenic mouse which expressed a GFP-tagged myelin-specific membrane protein[39] to generate Schwann cell–neuron myelinating cultures with neurons that expressed TdTomato (red fluorescence) in their cytoplasm. Time-lapse imaging of these cultures clearly showed that already at the initial (pre-myelination) Schwann cell–axon interaction, the Schwann cell sent thin processes that coiled around the axons that were to be myelinated (Fig. 2, Supplementary Section 1, Supplementary Fig. 1A, and Supplementary Movies 4 and 5). During the myelination process itself (Supplementary Fig. 1B and Supplementary Movie 6), we visualized slower and more prominent spiraling of the Schwann cell membrane around the axon, as demonstrated by a kymograph (Supplementary Fig. 1C), which probably represents wrapping of the inner myelin layer, as known to

occur during myelin formation[40] (also see Supplementary Fig. 1D, E, and Supplementary Movie 7). It is thus clear that coiling of Schwann cell membranes around axons is a main feature of the intercellular interactions between Schwann cells and the cylindrical axonal processes.

## Theoretical model

We consider a three-dimensional vesicle, which is described by a closed surface having $N$ vertices, each of them is connected to its neighbors with bonds, and forms a dynamically triangulated, self-avoiding network, with the topology of a sphere[30,41–43]. The vesicle is placed on a fiber, with which the vesicle has a uniform attractive contact-interaction (adhesion), as shown in Fig. 3. The initial condition of the vesicle may allow us to follow the dynamics of its spreading from an initial spherical shape that is placed within contact distance to the fiber, or from a state of pre-formed elongated protrusions that form on a flat substrate and are then mapped on to the fiber surface in an axial orientation (Fig. 5a). Note that our "minimal" model does not represent the whole cell, and does not contain a nucleus or other cellular components. It allows us to explore the dynamics of the membrane spreading on the membrane, with an emphasis on the dynamics of the leading edge of membrane protrusions.

The vesicle energy has the four following contributions: the bending energy is given by (integrated over the entire membrane area $A$),

$$W_b = \frac{\kappa}{2} \int_A (C_1 + C_2 - C_0)^2 dA, \qquad (1)$$

where $\kappa$ is the bending rigidity, $C_1$, $C_2$ are the two principal curvatures, and $C_0$ is the spontaneous curvature. We consider the spontaneous

curvature $C_0$ to be non-zero at the location of the curved proteins, while in the absence of any curved proteins, the spontaneous curvature is considered to be zero. We consider only convex curved proteins in our simulations. Note that the curved protein complexes in our model are isotropic, such that they have the shape of a spherical cap, without any in-plane orientation.

The direct binding energy between protein complexes residing on nearest-neighbor nodes, is given by,

$$W_d = -w \sum_{i<j} \mathcal{H}(r_0 - r_{ij}), \qquad (2)$$

where, $\mathcal{H}$ is the Heaviside step function, $r_{ij} = |\vec{r}_j - \vec{r}_i|$ is the distance between proteins, $\vec{r}_i, \vec{r}_j$ are the position vectors for $i, j - th$ proteins, and $r_0$ is the range of attraction, $w$ is the strength of attraction. The range of attraction is such that only the protein complexes that are on neighboring vertices can attract each other. The active energy is given by,

$$\Delta W_F = -F\,\hat{n}_i \cdot \vec{\Delta r}_i, \qquad (3)$$

where, $F$ is the magnitude of the active force, representing the protrusive force due to actin polymerization[44] that is acting in the direction of outward normal vector of the local membrane surface (along $\hat{n}_i$) and $\vec{\Delta r}_i$ is the displacement vector of the protein complex. The "active" forces in our simulations are implemented as external forces that act on specific nodes of the system. This is done by giving a negative energy contribution when the points on which these forces act move in the direction of the force. These forces are "active" since they give an effective energy (work) term that is unbounded from below and thereby drive the system out of equilibrium. By exerting a force directed at the outwards normal, we naturally describe Arp2/3-driven branching polymerization of actin, which forms actin filaments that grow in a wide range of angles, thereby resulting in an isotropic local pressure that is exerted on the membrane.

The above equation indicates that when the proteins are distributed inhomogeneously, there will be a net force on the vesicle in a particular direction[30]. However, in the present work, we only simulate vesicles that are adhered to fibers that are fixed in their location, which effectively links our vesicle to the lab frame, thereby restoring momentum conservation (fixing the fibers' position acts as an infinite momentum reservoir). In the cell, the polymerizing actin filament at the leading edge are linked by adhesion molecules to the fibers, thereby converting their polymerization dynamics into an effective protrusive force[45]. This process is implicitly assumed in our model, and is not described explicitly.

Finally, the adhesion energy is given by,

$$W_A = -\sum_{i'} E_{ad}, \qquad (4)$$

where $E_{ad}$ is the adhesion energy per adhered vertex, and the sum runs over all the vertices that are adhered to the fiber[30,41,46]. By "adhered vertices", we mean all such vertices, whose perpendicular distance from the surface of the fiber are less than $\epsilon$. We choose $\epsilon$ to be equal to the length $l_{min}$, which is the unit of length in our model, and defines a minimal length allowed for a bond. Thus, the total energy of the vesicle-fiber system is given by,

$$W = W_b + W_d + W_F + W_A \qquad (5)$$

We update the vesicle with mainly two moves, (1) vertex movement and (2) bond flip. In a vertex movement, a vertex is randomly chosen and attempt to move by a random length and direction, with the maximum possible distance restricted by $0.15 l_{min}$. In a bond flip move, a single bond is chosen, which is a common side of two neighboring triangles, and this bond is cut and reestablished between the other two unconnected vertices[30,41,46]. The maximum bond length is restricted to $l_{max} = 1.7 l_{min}$[41]. The choice of minimum and maximum bond length is to respect the self-avoidance of the triangulated network. We update the system using Metropolis algorithm, where any movement that increases the energy of the system by an amount $\Delta W$ occurs with rate $\exp(-\Delta W/k_B T)$, otherwise it occurs with rate unity.

Unless specified, we use a vesicle of total number of vertices, $N = 3127$ (radius ~ $20 l_{min}$). The bending rigidity $\kappa = 20 k_B T$, the nearest-neighbor protein–protein attraction strength $w = 1 k_B T$, and $\rho = N_c/N$ is the mean protein density, with $N_c$ vertices occupied by curved (convex) membrane proteins having spontaneous curvature: $C_0 = 1.0\, l_{min}^{-1}$. Note that we do not conserve the vesicle volume, although this could be maintained using an osmotic pressure term[41] or volume constraint. The membrane area is very well conserved ($\Delta A/A < 1\%$).

Note that our MC simulations do not describe the real dynamics of the fluid motion around the membrane and the fiber. However, it gives the energetically most favorable trajectory that controls the dynamics, without the correct timescales, which depend on the hydrodynamic dissipation. In addition, the actin filaments that exert the pushing forces at the leading edge of the protrusions, also transmit traction forces to the fiber, which can therefore move and bend it[25]. These traction forces are not explicitly described, as we maintain the fixed position of the fiber.

In this numerical study, we chose a fixed vesicle size such that we have a reasonable balance between accuracy and simulation time, for the range of fiber radii that we explored. The vesicle radius was chosen to be $R_{vesicle} \sim 20 l_{min}$, while the fiber radius was taken to be in the range of $3 - 11 l_{min}$. The mesh size for these dimensions allows us to describe the membrane shapes during the vesicle spreading process with good accuracy, even along the sharp (highly curved) leading edge of the protrusions. The leading edge curvature is determined by the spontaneous curvature of the curved proteins, which we kept from our previous work on vesicle spreading[30]. We quantified the accuracy by comparing the calculated mean curvature of the membrane with the discrete version, and find that even for a spherical membrane with a radius of a few $l_{min}$, the error is only ~10%. Clearly, membrane features that are on smaller length scales than $l_{min}$ are not captured by this calculation.

Note that our triangulated surface description of the membrane is valid for length scales that are larger than that of the membrane width or single-protein size. Our simulations are therefore valid for fibers that are larger than this length scale. Furthermore, we use the term "curved proteins" to describe any membrane-bound protein complex, or larger nano-domains, that have the properties of spontaneous curvature and ability to recruit the nucletion of actin polymerization. Let us emphasize that our continuum model of the membrane is not a microscopic model, where each node of the mesh represents a single protein or lipid.

The length scale $l_{min}$ that we use in our simulation does not necessarily correspond to be a particular length in reality. By assigning a different real length to the basic length scale of the simulation ($l_{min}$) we do not change the resulting dynamics, for the following reason: The dynamics of the spreading process depends on the energy gain/loss per additional adhered area increment (vertex). The adhesion energy per adhered vertex rescales as $1/l_{min}^2$, as does the bending energy per vertex (because the total bending energy of a specific shape is scale-invariant, provided we also rescale spontaneous curvature ($C_0$) accordingly). The active work is also independent of the absolute scale of $l_{min}$, as we define the force exerted by each vertex to be in units of $k_B T/l_{min}$, and therefore its contribution to the total energy increment per unit area, also scales as $1/l_{min}^2$. The absolute scaling of $l_{min}$ does not therefore change the qualitative nature of our simulation results: both the adhesion energy and the bending energy per unit area, scale as $1/l_{min}^2$.

The definitions of the different model parameters are given in Table 1. The choice of the parameters is based on our previous study[30], where we mapped the phases of spreading vesicles within this simplified model, and obtained the regime of parameters where protrusions with lamellipodia-like leading edge form. We therefore use here parameters identified in that previous study, so that we are able to simulate the dynamics of such protrusions on the fibers. The parameters of the model determine the magnitude of the different terms in the energy (Eq. (5)), such as the bending and adhesion energies. The

order of magnitude of the bending modulus $\kappa$ is well known for cellular membranes[47], and this gives the correct scale for the choice of all the other model parameters, such that their respective energy terms are comparable in magnitude. This interplay between different energy terms drives the complex dynamics in this system.

## Results

We first present the results for the spreading of a vesicle on a fiber with a circular cross-section. We study the steady-state shapes (Fig. 4), as a function of the fiber radius, and the average density of curved active proteins. This allows us to define the regime of parameters where our simulated vesicle spontaneously coils around the fiber. We then study the dynamics of this process by simulating an initially axially oriented vesicle, as it reorients spontaneously into a coiled shape (Fig. 5). Finally, we consider fibers with elliptical cross-section, exposing the critical aspect ratio above which coiling is inhibited in our simulations. These predictions are then compared with our experimental data (Fig. 6).

### Shapes of vesicles spreading on the fiber

We start by analyzing the dynamics of how a vesicle spreads on an adhesive fiber of a circular cross-section. The spreading of the vesicle over an adhesive cylindrical surface is determined by the balance between the bending and adhesion energies[30]. The curved membrane proteins, even when passive (do not recruit the protrusive forces due to actin polymerization, $F = 0$), can enhance the spreading

**Table 1 | List of parameters used in our simulation**

| Symbol | Definition | Unit |
|---|---|---|
| \multicolumn{3}{l}{Parameters used in simulation} | | |
| $N$ | Number of vertices | Number |
| $R$ | Radius of the round fiber | $l_{min}$ |
| $R_x(R_y)$ | Semi-major (minor) axis of the flat fiber | $l_{min}$ |
| $E_{ad}$ | Adhesion strength | $k_B T$ (per node) |
| $\rho$ | Average density of curved proteins | % |
| $w$ | Strength of protein–protein binding | $k_B T$ |
| $F$ | Active protrusive force | $k_B T / l_{min}$ |
| $C_0$ | Spontaneous curvature | $l_{min}^{-1}$ |

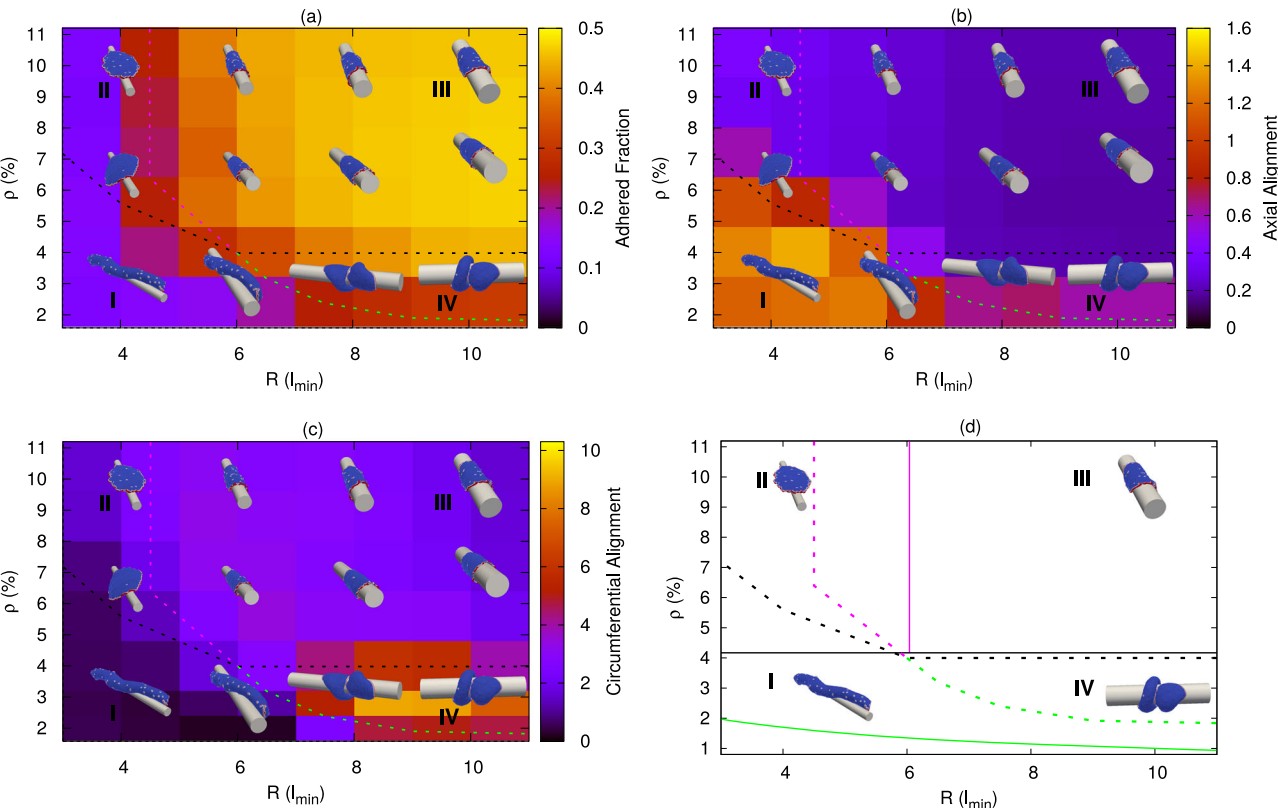

**Fig. 4 | Phase diagram of the steady-state vesicle shapes in the $R - \rho$ plane, where $R$ is the fiber radius and $\rho$ is the density of curved proteins.** The snapshots are shown for $R = 4, 6, 8, 10$ (in units of $l_{min}$), and $\rho = 3.2\%, 6.4\%$, and $9.6\%$. The red regions in the snapshots denote the curved proteins while the blue regions denote the bare membrane. Dashed lines are the transition lines generated from the simulation separating different phases. Here, we use adhesion strength ($E_{ad}$) $= 1.0 k_B T$, $F = 2.0 k_B T / l_{min}$. **a** The background color is showing the adhered area fraction of the vesicle, which is maximal for large $R$ and $\rho$ (phase III). **b** The background color is showing the variance of the

vertices locations along the cylinder axis (scaled by $10^3 \, l_{min}^2$), which is maximal for small $R$ and $\rho$, where the vesicle is aligned along the axis (phase I). **c** The background color is showing the angular variance of the vertices along the circumferential direction, which is maximal when the vesicle coils over the cylinder for large $R$ and small $\rho$ (phase IV). **d** Comparison between the transition lines between the different vesicle shapes from the simulations (dashed lines) and the analytical predictions (solid lines, with the equations given in Supplementary Section 3). Source data are provided as a Source Data file.

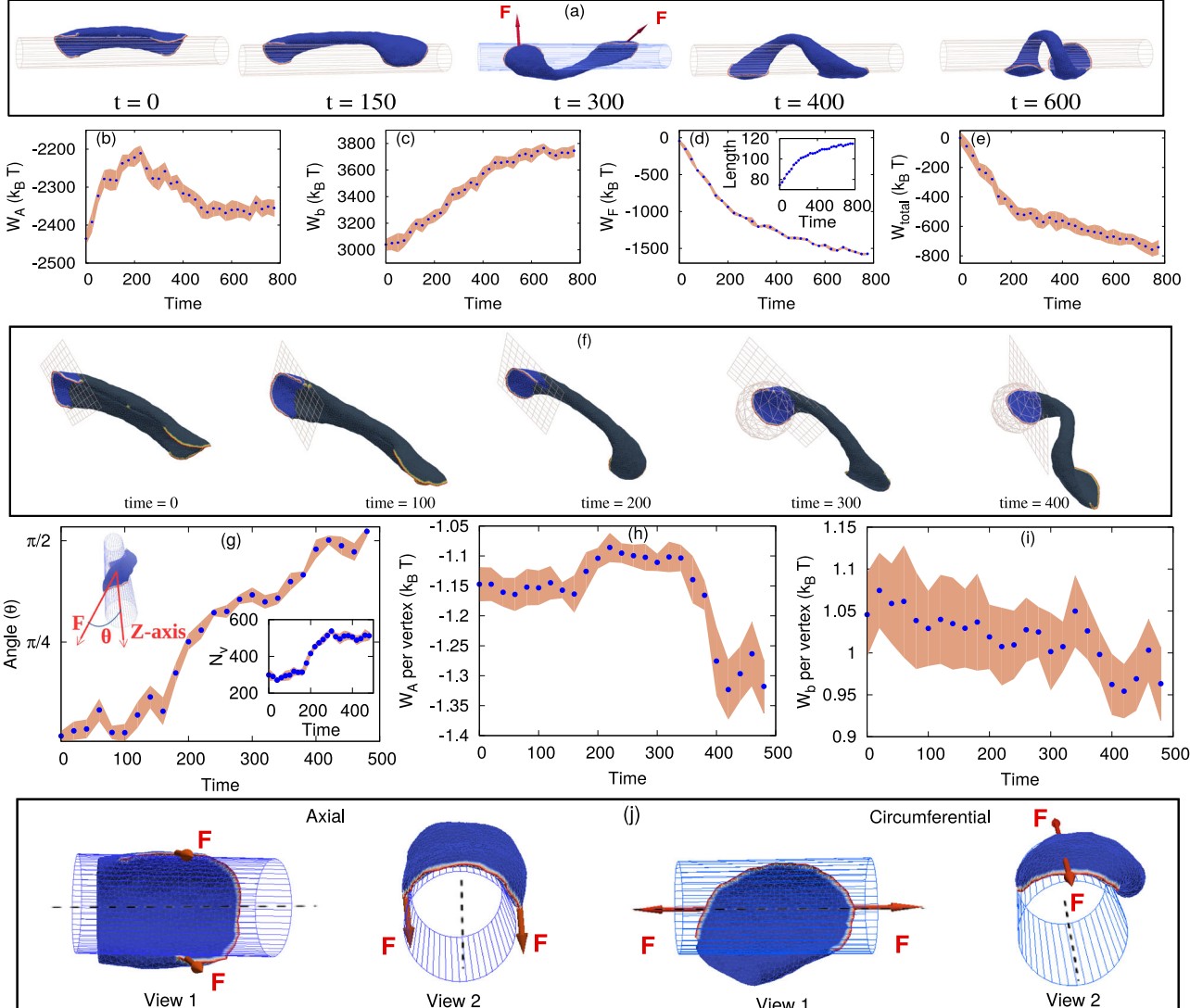

**Fig. 5 | Transition of a two-arc shape from axial to circumferential (coiling) orientation. a** Configurations of the vesicle at different times (in Monte–Carlo units). The red arrows on the 3rd inset are showing the direction of active forces acting on each of the two arcs of the elongated vesicle. **b** Variation of the total adhesion energy ($W_A$) with time, showing a non-monotonic variation. **c** The total bending energy ($W_b$) of the vesicle as a function of time, which is increasing monotonously during the coiling process. **d** Total work done by the active force ($W_F$), increasing the vesicle length over time (shown in the inset). The length is measured by assuming that the vesicle is organized like a helix, with the end positions being the centers of mass of the proteins in each of the two leading-edge arcs. **e** The total effective energy ($W_{total}$), sum of (**b**–**d**), which is a decreasing function of time. **f** Configurations showing the leading-edge region, as defined by the grid plane, and the grid-sphere at later times when the shape is highly coiled.

**g** The angle ($\theta$) between the direction of the total force acting at the cell edge (as denoted in **f**), with the cylindrical axis as a function of time. The inset shows the total number of vertices within this region. At longer times, the angle remains at $-\pi/2$. **h** Adhesion energy and **i** bending energy per vertex for the leading-edge region (as defined in **f**). **j** The configuration of an arc at the leading edge when it is oriented axially (as in $t = 0$ in **a**), versus circumferentially (as in $t = 600$ in **a**). The arrows denote the forces exerted by the leading-edge active proteins which are stretching the membrane of the leading-edge region axially, when the leading edge is oriented circumferentially, thereby increasing the adhesion energy of this membrane region (as shown in **h**). Here, we use $R = 10l_{min}$, $E_{ad} = 2.0k_BT$, $\rho = 2.4\%$ and $F = 2.0k_BT/l_{min}$. The unit of time in the plots is normalized to be $2 \times 10^4$ MC steps. **b**–**e**, **g**–**i** Data are presented as mean values ± SD. Source data are provided as a Source Data file.

by reducing the bending energy cost. This is shown in Supplementary Section 2, Supplementary Figs. 2 and 3 (Supplementary Movies 8–12), with a monotonously increasing adhered area as the adhesion energy, radius of the cylindrical fiber ($R$) and the average density of curved proteins ($\rho$) increase. These systems do not exhibit any tendency for rotations or coiling dynamics.

In Fig. 4, we describe the steady-state shapes for vesicles with active curved proteins ($F \neq 0$), as a function of $R$ and $\rho$. We notice that there are several distinct phases of adhered vesicles on the fiber, which are marked by the colored "transition" lines (summarized in Fig. 4d). The dashed lines denote the transition lines extracted from the simulations, as explained below, while the solid lines give an approximate

analytic calculation of the transition lines (given in the Supplementary Section 3).

In both phases (II) and (III), the steady-state vesicle is in a flat, pancake-like shape. This shape was found in previous works using this model, both without any adhesion[41] and when spreading over a flat substrate[30]. It occurs when the density $\rho$ is high enough to allow the curved proteins to form a closed circular cluster around the rim of the flat vesicle. The critical line below which the density is too low to allow the pancake shape to exist, is denoted by the dashed black line. Phases (I) and (IV) exist below this transition line.

Phase (I) occurs for small $R$ and $\rho$, the vesicle shape is elongated (aligned axially, along the long axis of the fiber), with a very small

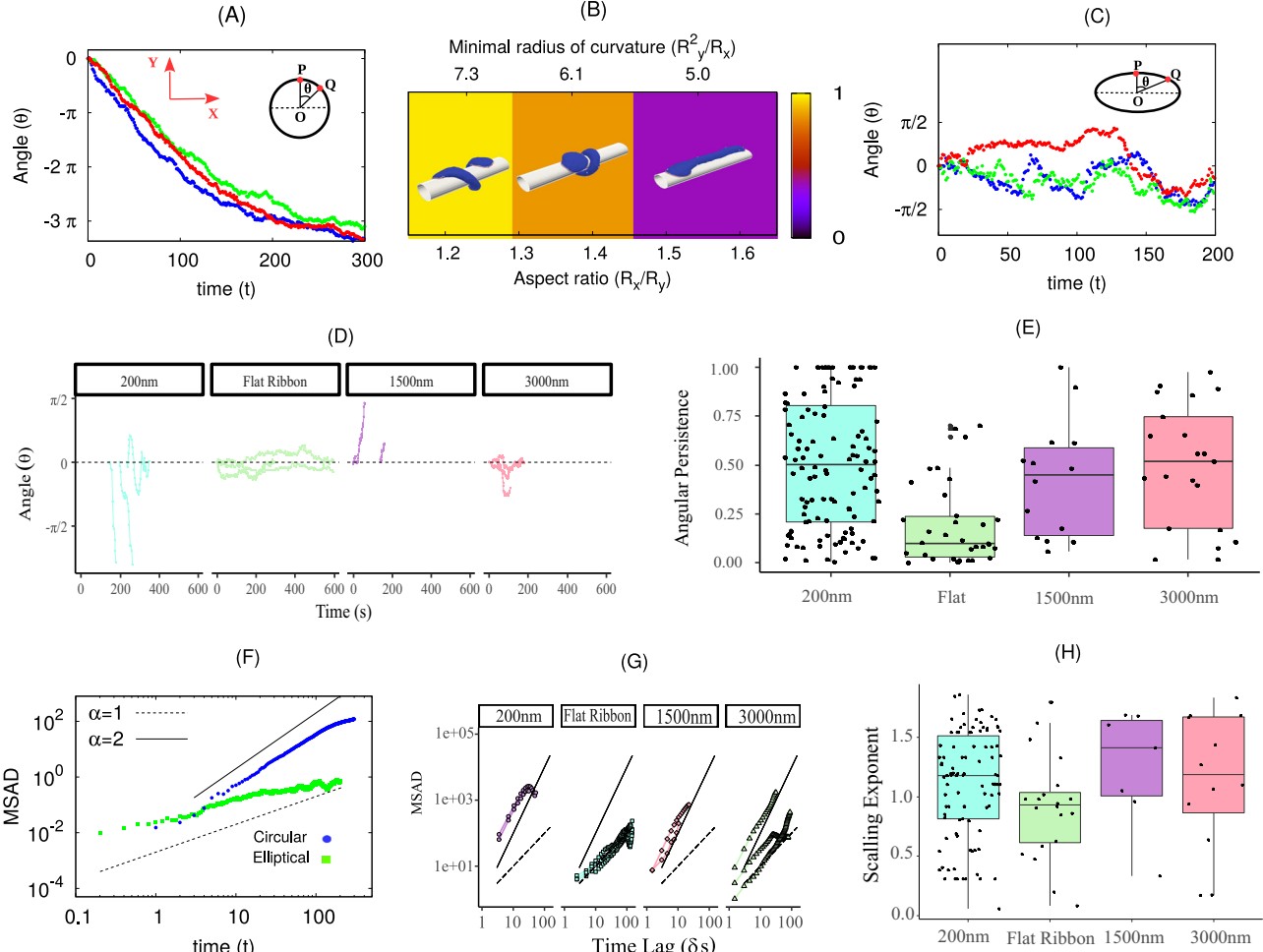

**Fig. 6 | Membrane coiling dynamics on fibers of circular and elliptical cross-section s. A** Simulated angular displacement ($\theta$) of the leading edge of vesicle with time for the circular fiber ($R = 10l_{min}$). Different colors represent different realizations. The inset shows the definition of angular displacement $\theta$. Here, $P$ and $Q$ are the initial and final position of a leading-edge protein on the $X-Y$ plane, and $\theta$ is the angular displacement between them. We use here $E_{ad} = 1.0k_BT$, $\rho = 2.4\%$ and $F = 2.0k_BT/l_{min}$. The unit of time is $10^5$ MC steps. **B** Typical configurations of the simulated vesicles on fibers with elliptical cross-sections. The background color gives a quantification of the circumferential alignment of the vesicle, as in Fig. 4c. The coiling ceases as the aspect ratio of the elliptical cross-section ($R_x/R_y$) increases above -1.6. The circumference length of the elliptical cross-section is kept constant, equal to the circumference of the circular cross-section with $R = 10.0l_{min}$. **C** Simulated angular displacement of the leading edge of the vesicle as function of time, for an elliptical fiber of aspect ratio 1.6. Different colors represent different realizations. **D** Experimental data for the angular position with time for different types of fibers, using C2C12 cells on polystyrene fibers coated with Fibronectin (as in Fig. 1A). **E** Experimental results for angular persistence for various types of circular and flat fibers. Statistics for the boxplot are, 200 nm: 0.01 minima, 0.37 median, 0.94 maxima, the number of independent experiments $n = 65$; Flat ribbon: 0.01 minima, 0.10 median, 0.70 maxima, $n = 29$; 1500 nm: 0.11 minima, 0.34 median, 0.9 maxima, $n = 10$; 3000 nm: 0.02 minima, 0.43 median, 0.89 maxima, $n = 16$. **F** Mean square angular displacement (MSAD) from simulations, for circular and elliptical (aspect ratio -1.6) fibers. The initial growth of MSAD is $-t^\alpha$, where $\alpha = 1$ represents diffusive behavior while $\alpha = 2$ shows ballistic nature. **G** MSAD from the experiments, on fibers of various cross-sections. **H** The value of the power-law exponent ($\alpha$) from the experimental MSAD (**G**). Statistics for the boxplot are, 200 nm: -0.89 minima, 1.19 median, 1.91 maxima, $n = 65$; Flat ribbon: -1.82 minima, 0.74 median, 1.80 maxima, $n = 29$; 1500 nm: −0.03 minima, 1.23 median, 1.69 maxima, $n = 10$; 3000 nm: −0.04 minima, 0.43 median, 0.89 maxima, $n = 16$. Source data are provided as a Source Data file.

adhered area. Since $R$ is small, and there are not enough curved proteins to form a ring-like aggregate around the whole rim of the vesicle (as in phase II, below), the vesicle can only adhere axially and in this way minimize its bending energy. We identify this phase as the region of the phase diagram where the axial elongation of the adhered vesicle is maximal (Fig. 4b and Supplementary Movie 13) as measured by the variance in the distribution of vertices along the axial direction ($Z$ axis) (background color in Fig. 4b, and Supplementary Section 4, and Supplementary Fig. 5). This axial extension of the vesicle allows us to define a contour that separates phase (I) from the coiled phase (IV), as denoted by the dashed green line.

Phase (II) occurs in the regime of small $R$, above the critical $\rho$ to form a flat pancake-like shape with all the proteins clustered as a ring around the rim. For small $R$, the bending energy cost of the pancake-

like vesicle for wrapping around the fiber is too high, and it remains "hanging" on the fiber[15] (Supplementary Movie 14).

In phase (III), the larger value of $R$ allows the vesicle to fully adhere, as the adhesion energy gain now dominates over the (much smaller) bending energy cost of wrapping around the fiber. This phase is identified by having the maximal adhered area fraction of -0.50 (background color in Fig. 4a and Supplementary Movie 15). We therefore use a contour of the adhered area fraction to define the transition line between phases (II) and (III), denoted by the dashed pink line.

Finally, phase (IV), occurs at large $R$, for $\rho$ below the critical value for the formation of the flat (pancake-like) shape. In this phase vesicles tend to form a two-arc phase[30,41], where two aggregates of proteins form at opposing ends of the cell, stretching the elongated membrane

between them. This shape spontaneously orients to point along the circumferential direction, and pull the vesicle into a coiled helical structure. This phase is therefore identified by the large overall angular spread of the vesicle along its length (Fig. 4c and Supplementary Movie 16), quantified by the variance in the angular distribution of the vertices along the circumferential direction (background color in Fig. 4c and Supplementary Section 4, and Supplementary Fig. 6). The lines that form the boundary of this region, which is identified by large angular coiling, coincide with the transition lines already denoted by the black and green dashed lines.

Note that in the parameter regime of phase IV (coiling phase), the vesicles can also form a phenotype where all the proteins are aggregated in a single cluster. These vesicles become motile, as observed on a flat adhesive substrate[30].

### The mechanism driving the coiling phase

We now analyze the coiling phase (IV, Fig. 4), to expose the mechanism that drives the circumferential orientation of the leading edges of the membrane protrusions.

We consider a two-arc vesicle, generated on a flat substrate, and place it on the cylindrical fiber along the axial orientation as shown in Fig. 5a. We find that the vesicle is unstable in the axially aligned state, and the two leading edges spontaneously rotate to circumferential alignment, thus causing the vesicle to coil (Supplementary Movie 17). We follow the evolution of the energy components during this process (Fig. 5b–e). Globally, the adhesion energy ($W_A$) increases at the early stage (Fig. 5b, time $\lesssim 200$), as the tubular middle part of the vesicle partially detaches from the curved fiber surface and is stretched by the active force (Fig. 5d, inset). The overall bending energy ($W_b$) increases throughout the process, as the tubular part is stretched to become thinner and is bent (coiled) circumferentially around the fiber (Fig. 5c).

These energy penalties are counter-balanced by the work done by the active force ($W_F$) in stretching the vesicle (Fig. 5d). This energy is calculated as the integrated change in length of the vesicle multiplied by the net active force component that is aligned along the stretch direction (red arrows in Fig. 5a at $t = 300$). As was shown for the two-arc configuration on a flat substrate[30], the active work contributes a negative term to the total effective energy of this configuration (Fig. 5e), which is the sum $W_{total} = W_A + W_b + W_F$, and acts to stabilize it. Note that protein–protein binding energy ($W_d$) does not have a significant contribution during this coiling process, as the clusters of proteins at the two leading edges remain almost constant in size (see Supplementary Section 5 and Supplementary Fig. 7). At a later stage (time $\geq 200$, Fig. 5b), the adhesion energy is partially recovered as the flat leading edges are stretched along the axial directions by the active forces, increasing their adhered area (Fig. 5j, h). Overall, the total effective energy decreases (becomes more negative) over time (Fig. 5e), so the process continues until full coiling.

Note that there is a low probability for the two arcs of the initially axial vesicle (Fig. 5a, $t = 0$) to reorient in the same direction along the circumference. In such a case, the two arcs will merge to form a motile crescent vesicle[43,48].

While the coiling process is therefore mainly driven by the active forces that elongate the vesicle, performing active work, the above analysis does not explain the origin of the initial reorientation of the leading edges from the axial to the circumferential alignment. In order to understand this reorientation process, we need to "zoom-in" on the dynamics of the leading edges (during the process shown in Fig. 5a). We consider a section of the vesicle which contains the leading-edge protein aggregate and the flat membrane protrusion that it forms (Fig. 5f). We define this leading-edge region as follows: we draw a plane perpendicular to the direction of the net force of each protein arc, and place it in a position such that all the proteins forming that arc are on one side of this plane. This criterion is not sufficient when the shape of

the arc is highly coiled, as it accepts nodes that are on the other side of the coiled shape. We therefore also use another constraint (Fig. 5f at time $\geq 300$), by considering the leading-edge region for all nodes that are within a distance $r_{max}$ from the center-of-mass (COM) of the proteins forming the arc, where $r_{max}$ is the maximum distance of a protein in the arc from the COM of the proteins. Together, these criteria define a leading-edge membrane region that slightly fluctuates in size over time (Fig. 5g, inset, where we see that the biggest change in the number of vertices occurs at time ~300).

At the beginning of the process, the total force due to the proteins in each leading-edge arc is directed along the axial direction, and the angle of this force with the cylinder's axis increases until it is perpendicular (~$\pi/2$) at later times (Fig. 5g), when the coiling reaches its stationary state (Fig. 5a). The adhesion and bending energies per vertex of the membrane within the leading-edge region are shown in Fig. 5h, i. We plot the energies per vertex to remove the effect of the variation in the number of vertices over time (Fig. 5g, inset). The adhesion energy fluctuates in the beginning but there is an overall decrease, while the bending energy slightly decreases throughout the process. The total value of adhesion and bending energies without scaling by the number of vertices are shown in Supplementary Section 6 and Supplementary Fig. 8.

These changes arise from the flat membrane at the leading-edge region being more stretched along the axis (the zero curvature direction) by the active proteins when oriented circumferentially, thereby increasing the contact area and decreasing both the adhesion and bending energy (Fig. 5j). In the initial axial orientation, the active forces of the proteins along the leading-edge are not as effective in stretching the membrane sideways, as this involves strong bending of the membrane around the fiber (Fig. 5j), and therefore encounters a large bending energy penalty. From these observations, we conclude that the reorientation process of the leading-edge regions is mainly driven by locally decreasing the adhesion and bending energies, while the global coiling process of the whole vesicle is driven by the work done by the active forces (Fig. 5e).

### Theoretical predictions compared with experiments

We now use our theoretical model to make several predictions that we then test in experiments.

In Fig. 6A, we plot the angular displacement ($\theta$) of the proteins on the leading edges of the membrane protrusions, where $\theta$ is defined as the angle between the initial and subsequent location of a leading-edge protein (measured from the center of the fiber) on the $X-Y$ plane. We next vary the cross-section shape of the fiber by considering an elliptical cross-section having the same circumference as the circular fiber with a radius $R = 10 l_{min}$. We find that above a critical aspect ratio (ratio of the semi-major to the semi-minor axis of the ellipse ~1.6), the vesicle does not coil but rather remains axially aligned (Fig. 6B and Supplementary Movie 18). The minimal radius of curvature for this cross-section is $5 l_{min}$, which is a little bit smaller than the radius at which coiling stops for a fiber with a circular cross-section ($R \lesssim 7 l_{min}$ in Fig. 4). The coiling dynamics of the leading edges of the vesicle on a fiber with such a high aspect ratio is inhibited, and the value of the angular position of the leading-edge fluctuates around zero (Fig. 6C), while on fibers with circular cross-section the angular displacement of the leading edges increases beyond $|2\pi|$, and saturates when the vesicle is fully coiled (Fig. 6A at t >200).

We verify this prediction using our experimental observations of the trajectories of membrane ruffles on round and flat artificial fibers (experimental images and extracted trajectories are shown in Fig. 7; details of the experimental methods are in "Methods"). Note that in the simulations, we model the coiling of the leading edge that is adhered to the fiber surface, while in the experiments we could visualize the motion of membrane ruffles that extend above the fiber surface. Nevertheless, the motion of these ruffles appears to be correlated with

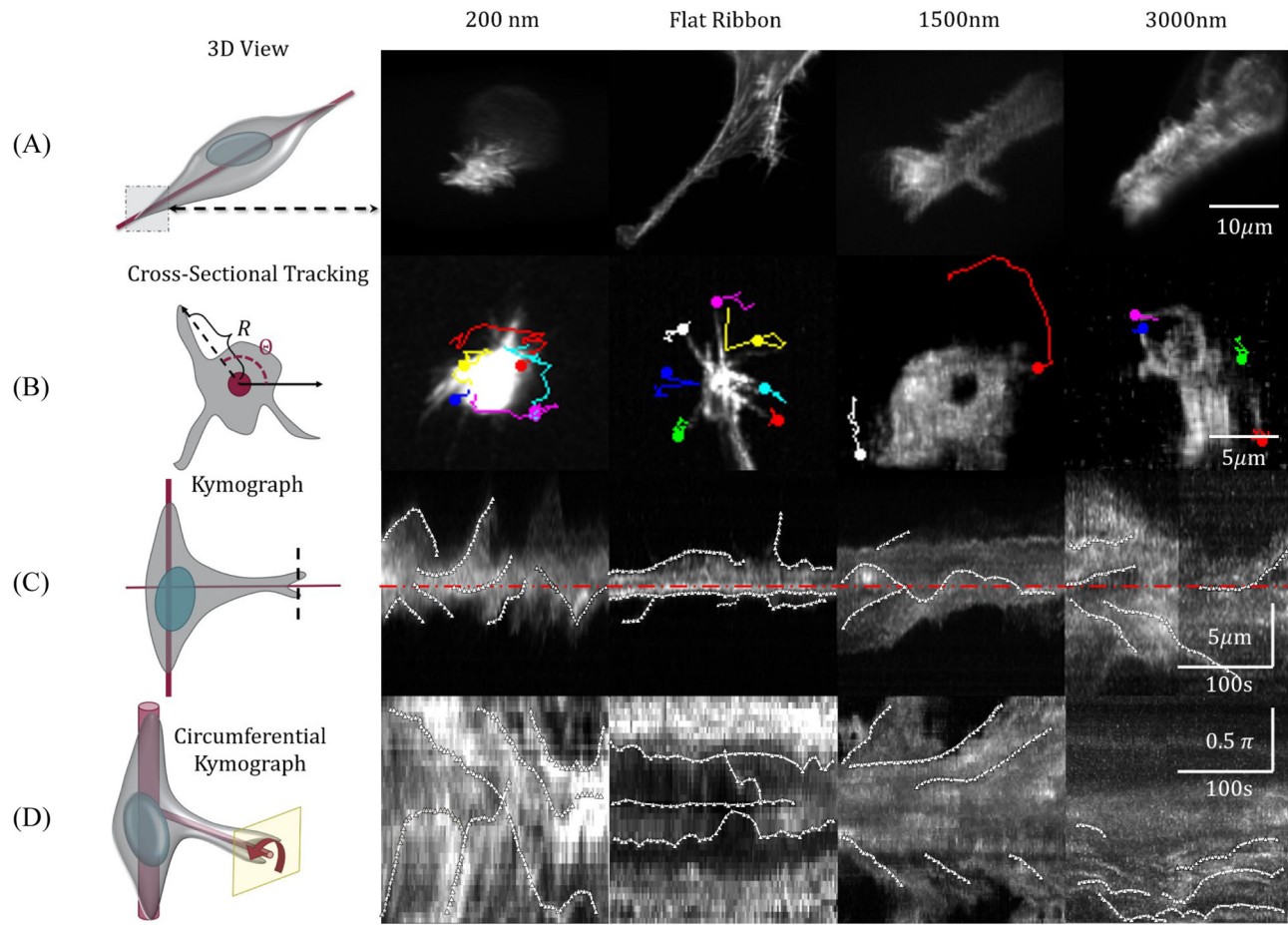

**Fig. 7 | Coiling at the leading edges of cellular protrusions, on fibers of different diameters, using C2C12 cells on polystyrene fibers coated with Fibronectin (as in Fig. 1A).** The round polystyrene fibers have a diameter of 200, 1500, and 3000 nm, while the "Flat Ribbon" fibers are produced by flattening a 200-nm fiber, increasing its cross-sectional width to $-\pi D \simeq 600$ nm with a thickness $\simeq 50$ nm. The left panels give a schematic illustration of the cell (gray region), spread over the fiber(s) (red solid lines), with the nucleus indicated by the blue oval. **A** Maximum intensity projection of the inclined volumetric scan of the cells on the fibers, with Dual-View Light Sheet microscopy, as illustrated by the representative cartoon on the left. **B** The cartoon on the left establishes the polar coordinates used for the tracking of the ruffles. The following images illustrate representative coiling trajectories of membrane ruffles. The colored lines trace the displacements of the ruffles, with trajectories ending in a circle. **C** Kymographs along a perpendicular dashed line located near the leading edge, as shown in the cartoon, for the same cases as in (**B**). These illustrate the movement of the ruffles with respect to the axis of the fiber (denoted by the red dashed center line). Kymographs qualitatively and quantitatively show how the actin-filled ruffles are able to rotate around the fibers, except around the flat ribbons. **D** Circumferential kymographs made along a circular path around the axis of the fiber, in a slice located near the leading edge, as shown in the schematic drawing. These kymographs show the motion of the membrane with respect to the circumferential position around the fiber. The white traces indicate rotations of actin-filled ruffles around the fibers, except for flat ribbons. Source data are provided as a Source Data file.

the coiling motion of the underlying leading-edge membrane[27] (as discussed in "Introduction" in relation to Fig. 1).

We find that on a fiber with a round cross-section (diameter 200 nm) the ruffles exhibit highly directed angular trajectories (Fig. 6D), while on the flat-ribbon fibers the trajectories fluctuate around zero (see corresponding trajectories on the cell images in Fig. 7B). The experimental trajectories are noisy, but are qualitatively in close agreement with the theoretical simulations (Fig. 6A, C). The directed trajectories persist for fibers of larger diameter (1500 nm), but lose their directionality when the fiber diameter is even larger (3000 nm). These differences in the experimental trajectories can also be quantified by their angular persistence (Fig. 6E), defined as the ratio of the net angular displacement of the coiling event divided by its cumulative angular displacement along the trajectory.

The difference between coiling dynamics on round and flat fibers can be further quantified by plotting the mean square angular displacement (MSAD) for both cases. The MSAD is expected to vary as $-t^{\alpha}$, where $\alpha = 1$ for a diffusive behavior, while $\alpha = 2$ represents ballistic, persistent coiling. In the simulations, we find ballistic behavior for round fibers (before the coiling saturates due to finite membrane area), while diffusive for fibers with a highly elliptic cross-section (Fig. 6F). The experimental data exhibits the same trends (Fig. 6G), with ballistic dynamics on the round fibers (of diameter $D = 200$ nm and 1500 nm), and diffusive motion on the flat-ribbon fiber. On the round fibers with the largest diameter (3000 nm), we observe mixed behavior. The values of the exponent $\alpha$, extracted from the experimental data, are summarized in Fig. 6H. The coiling velocity was found to decrease slightly as the fiber radius increases, which is not observed in our simulations (shown in Supplementary Section 7 and Supplementary Fig. 9).

The experimental images of the cells' leading edges and the trajectories of the ruffles are shown in Fig. 7 (also see Supplementary Movies 19–23). Figure 7A shows the isometric view of the cells' protrusions on the fibers. The ruffles are highly dynamic, with large coiling motion, for the round fibers, while significantly less for the flat-ribbon fiber (Fig. 7B). The kymographs in Fig. 7C, D clearly demonstrate that the ruffles on the round fibers coil around the fiber axis, while on the flat-ribbon fibers they stay only on one side and do not wrap around the axis.

In Supplementary Fig. 10A, B, we show that reducing the adhesion strength below a critical value in our model causes the collapse of the leading-edge clusters, and the vesicle does not coil. The result is that the vesicle spreads along the fiber axis, similar to regime I in Fig. 4. Experimentally, coiling was observed for low concentration of fibronectin coating, and even in its complete absence, since cells apparently secrete enough coating to allow them to adhere strongly to the plastic fibers (Supplementary Fig. 10C–E and Supplementary Movie 24). We do find however that the persistence of coiling events was reduced when the adhesion strength decreased, while at high levels of adhesion, the ruffles become more strongly adsorbed to the fiber, making it very difficult to accurately follow their coiling motion in the experiment. These results further demonstrate the robustness of the coiling phenomenon.

## Discussion

Cells often encounter extracellular fibers on which they adhere, spread and migrate. We show that the tendency of the leading edge of cellular protrusions to coil around adhesive fibers can be understood using a physical model with a minimal set of components: membrane with curved membrane protein complexes that induce outwards active forces (representing the recruitment of cytoskeletal activity), and adhesion. Within this model, the coiling process of membrane protrusions adhering to the fibers emerges spontaneously, and is driven by the physics of minimizing the free energy and the active work. This physics-based model predicts that the coiling will be inhibited when the radius of curvature of the fiber is too small, due to high membrane bending energy. This prediction is verified in our experiments, comparing fibers with circular cross-section and flattened fibers. Furthermore, the essential role of the curved membrane proteins in our model can explain the reduced coiling observed in cells with reduced amount of IRSp53 proteins[49], which have convex curvature, and are involved with the recruitment of Arp2/3 actin nucleation to the membrane.

Our theoretical work indicates that there is a fundamental physical mechanism that leads to coiling dynamics of membrane protrusions, which can then be further tuned and regulated by biological signaling. The model therefore describes a general mechanism, that can explain the tendency of coiling dynamics at the leading edge of membrane protrusions, in cells of different types, such as Mouse Muscle Myoblasts on artificial fibers and glial cells extending over axons (Fig. 2). Our model therefore unifies these observations of coiling from different biological systems. Furthermore, myelination is known to occur on axons for which the diameter exceeds a critical threshold[35]. This theoretical model reveals a physical mechanism that inhibits coiling, and may correspondingly prevent myelination around very thin fibers in the vertebrate nervous system.

We note that our model is highly simplified, and does not explicitly includes many components that could affect the coiling dynamics. These include, for example, the elastic rigidity of the cortical actin, which may effectively greatly increase the bending rigidity of the membrane protrusions, therefore resisting the coiling dynamics. In addition, there may be bundled actin nucleated at the protrusion leading edge, in addition to the branched actin that our model more naturally describes. Such bundled actin induces filopodia growth, which are observed in cellular protrusions (such as neuronal growth cones[50]) and may inhibit the ability of the lamellipodia-like leading edge to induce coiling. Finally, it has to be remembered that our simulations describe a simplified membrane protrusion, while cells spreading over fibers have to accommodate internals organelles, such as the nucleus, and contain cytoskeletal structures, such as stress-fibers[49], which can further modify the coiling behavior.

In conclusion, we propose a physics-based mechanism for the tendency of adhering cellular membranes to coil around fibers. The mechanism is driven by the out-of-equilibrium (active) nature of the forces exerted by the cytoskeleton on the membrane, and the feedback between curved membrane proteins, adhesion and membrane bending. This mechanism may underlie crucial processes in biology, and remains to be systematically tested by more detailed molecular studies.

## Methods

### Imaging of the coiling of leading-edge protrusions on suspended fibers

**Scaffold preparation.** Using the previously reported non-electrospinning STEP technique[51–53], suspended fiber nanonets composed of 200 or 500 nm diameter fibers spaced 20 μm apart were deposited orthogonally on 2 μm diameter fibers spaced 300 μm apart. Nanofibers were manufactured from solutions of polystyrene (MW: 2,500,000 g/mol; Category No. 1025; Scientific Polymer Products, Ontario, NY, USA) dissolved in xylene (X5-500; Thermo Fisher Scientific, Waltham, MA, USA) in 6 and 9 wt% solutions. 1- and 3.5-μm-diameter fibers were manufactured from 2 and 5 wt% of high molecular weight polystyrene (MW: 15,000,000 g/mol, Agilent Technologies, Santa Clara, CA, USA) and equally spaced at ~60 μm. The polymeric solutions were extruded through micropipettes with an inside diameter of 100 μm (Jensen Global, Santa Barbara, CA, USA) for deposition of fibers on a hollow substrate. All fiber networks were cross-linked at intersection points using a custom fusing chamber, to create fixed-fixed boundary conditions. 200-nm diameter fibers were made into flat ribbons (of width πD, where D is the diameter of the original fiber of circular cross-section) as described in ref. 15.

**Cell culturing.** Mouse Muscle Myoblasts (C2C12, ATCC) expressing GFP actin were cultured on petri dishes using Dulbecco's Modified Eagle Medium (DMEM, Gibco, Thermo Fisher Scientific) with 10% fetal bovine serum (FBS, Invitrogen, Carlsbad, CA, USA) and 1% penicillin–streptomycin. Lentivirus was generated in HEK293T cell line using calcium transfection method with VSV.G and psPAX2 packaging plasmids (Addgene #14888 and #12260) and pLenti.PGK.LifeAct-GFP.W (Addgene #51010). Harvested lentiviral supernatant was concentrated 100 × by centrifugation at 50,000 × g for 2 h at 4 °C. For transduction, 100,000 cells were seeded in a six-well plate. The following day, media was removed from cells and transduction media consisting of 150 μl concentrated virus + 650 μl of media (DMEM + 10% FBS) + 8 μg/ml polybrene (Millipore Sigma, TR-1003-G) was added dropwise onto cells. Virus was removed the day after. Finally, GFP-positive cells were collected on the Sony SH800 cell sorter at the JHU Integrated Imaging Center, expanded, and frozen in 5% DMSO. STEP-spun scaffolds were placed on 100 × 20-mm Petri dishes and disinfected with 70% ethanol, then coated with 6 μg/ml of Rhodamine Fibronectin (Cat. # FNR01, CYTOSKELETON, Denver, CO, USA) by incubating at 37 °C for 1 h. The trypsinized and resuspended cells were seeded onto STEP-spun scaffolds, allowed to attach for at least 2 h and the wells were flooded with 2 mL of DMEM + 10% FBS.

**Cell imaging with dual-view light-sheet microscopy.** The seeded samples with DMEM were PBS washed, then flooded with Live-Cell Imaging Solution (Invitrogen, Carlsbad, CA, USA) for imaging using Dual-View Plane Illumination Microscopy (diSPIM)[33,34]. Samples were mounted in the diSPIM and kept at 37 °C after calibrating the light-sheet movement to the piezo step factor using a micromanager plugin[54]. Cells were volumetrically imaged (two views), up to 120 time points at 1.5, 2.5, and 3.5-s intervals constrained by the cell size, the number of slices, and location in the suspended nanonets.

The diSPIM configuration used in these experiments is a minor modification of that described in detail elsewhere[33,34]. Two 40× 0.8NA Nikon CFI APO NIR objectives were used for illumination and imaging that, upon joint deconvolution of the two acquired views, provide an isotropic resolution of 330 nm. Excitation was provided by an OBIS 488 nm LX 150 mW laser and an OBIS 561-nm LS 150 mW laser

(Coherent) coupled to the microscope via optical fibers. Green and red fluorescence was excited and collected simultaneously and split onto the camera chips using W-VIEW GEMINI Image splitters (Hamamatsu) equipped with a 561-nm long-pass dichroic mirror, a 525/50-nm bandpass emission filter, and a 568-nm long-pass edge filter (all Semrock). Fluorescence was imaged with ORCA Fusion BT CMOS cameras (Hamamatsu).

The microscope was controlled by the DISPIM plugin in Micromanager[55]. Typical image stacks were acquired with a 1 ms camera exposure time, a slice step of 0.5 μm, over 50–120 slices, with a the total imaging field spanning a volume ranging from $41.6 \times 332.8 \times 25\,\mu m^3$ to $170.3 \times 332.8 \times 60\,\mu m^3$ (FOV of $2048 \times 256$-$1048$, width × height, respectively). The power at the sample ranged from 700 μW to 2 mW.

Z stacks acquired from the diSPIM were background subtracted, bleach corrected, cropped, and sorted in Fiji[56]. The orthogonal views (SPIM A and SPIM B) were fused using a GPU-optimized pipeline as described in Guo 2020[57]. Fused stacks with isotropic resolution were rotated with the TransformJ plugin for visualization and analysis. Maximum intensity projections derived from the volumetric time series were used to manually track the protrusive activity at the coiling fronts of cells. RStudio was used to generate all plots and statistical comparisons of means using the Kruskal–Wallis test.

### Imaging of early-stage myelination process

For the generation of Schwann cell-dorsal root ganglia (DRG) neuron myelinating cultures, we used mice expressing S-MAG-GFP (a transgene expressed specifically in myelinating cell membranes[39]). We used a pregnant female at E13.5 days of gestation. The mouse was on C57/BL6 background, carrying an S-MAG-GFP transgene under the MAG promoter. The mouse was generated by Nicole Schaeren-Wiemers from the Department of Biomedicine, University of Basel. DRG cultures were prepared from mouse embryos at day 13.5 of gestation. DRGs were dissociated and plated at a density of $4 \times 10^4$ per chamber (Lab-Tek coverglass system Nunc #155411), coated with Matrigel (Becton Dickinson) and poly-D-lysine. Cultures were grown for 2 days in Neurobasal medium supplemented with B-27, glutamax, penicillin /streptomycin and 50 ng/ml NGF. At day 2 post seeding cultures were infected with a lentivirus carrying a cytoplasmic Tdtomato reporter, resulting in neuronal expression of TdTomato. Cultures were then grown for 4 additional days in BN medium containing Basal medium-Eagle, ITS supplement, glutamax, 0.2% BSA, 4 mg/ml D-glucose, 50 ng/ml NGF, and antibiotics. To induce myelination, cultures were grown in BNC, namely a BN medium supplemented with 15% heat-inactivated fetal calf serum (replacing the BSA) and 50 μg/ml L-ascorbic acid.

Cultures were imaged at 7 days with myelinating medium (for fluorescence imaging, at a frequency of four frames per hour for 65 or 19 h). Fluorescence images were obtained using a confocal microscope (LSM700 confocal microscope Carl Zeiss) 488-nm and 555-nm laser lines. Confocal time-lapse images were captured using a Plan-Apochromat 20 × /0.8 M27 objective (Carl Zeiss), at temperature, CO2 and humidity-controlled conditions. Image capture was performed using acquisition Blocks (Carl Zeiss Zen 2012). Images and movies were generated and analyzed using Zeiss Zen 2012 and Adobe Photoshop CC 2019. All experiments were performed in compliance with the relevant laws and institutional guidelines and were approved by the Weizmann Institute's Animal Care and Use Committee.

### Reporting summary

Further information on research design is available in the Nature Portfolio Reporting Summary linked to this article.

## Data availability

All the source data to generate the figures are provided with this paper. Source data are provided with this paper.

## Code availability

All code for conducting simulations with configuration files for acquisition of the results associated with the current submission is openly available at GitHub https://github.com/rajsadhu3903/Coiling-codes. Any future updates will also be published in the same GitHub repository.

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

## Acknowledgements

A.S.N. acknowledges partial funding support from the National Science Foundation (NSF, Grant Nos. 1762634, 2119949, and 2107332). The opi-nions, findings, and conclusions, or recommendations expressed are those of the author(s) and do not necessarily reflect the views of the National Science Foundation. A.S.N. and B.B. acknowledge the Institute of Critical Technologies and Science (ICTAS) and Macromolecules Innovative Institute (MII) at Virginia Tech for their support in conducting this study. H.S. acknowledges the support from the intramural research program of the National Institute of Biomedical Imaging and Bioengi-neering within the National Institutes of Health. A.I. and S.P. acknowl-edge the support from the Slovenian Research Agency (ARIS) through Programme No. P2-0232 and projects Nos. J3-3066 and J2-4447. N.S.G. is the incumbent of the Lee and William Abramowitz Professorial Chair of Biophysics, and acknowledges support by the Ben May Center for The-ory and Computation, and the Israel Science Foundation (Grant No. 207/22). N.S.G. acknowledges the support of the Ilse Katz Institute for Material Sciences and Magnetic Resonance Research. E.P acknowl-edges the Israel Science Foundation, Dr. Miriam and Sheldon G. Adelson Medical Research Foundation, and Estate of Paula Vial Lempert, Sassoon and Marjorie Peress, the Estate of Gerda Wassermann, and the Estate of Dr. Sylvia Brody Axelrad. This research is made possible in part by the historic generosity of the Harold Perlman Family.

## Author contributions

R.K.S., A.I., S.P., and N.S.G. developed the theoretical model; R.K.S. and N.S.G. conceived, designed, and implemented the analysis of the

model, and prepared the manuscript; A.S.N. and N.S.G. conceived and supervised the project; C.H.P, L.Z., H.D.V, B.B., K.K., H.S., and A.S.N. conceived and supervised the experiments on suspended fibers and contributed the data; E.P. and Y.E.E. performed the experiments and analysis of the myelination process and contributed the data. All authors reviewed and edited the manuscript.

## Competing interests

The authors declare no competing interests.
