## [Peer Review file · Nature Communications]

Experimental and Theoretical model for the origin of coiling of cellular protrusions around fibersREVIEWER COMMENTS

Reviewer #1 (Remarks to the Author):

Cells migrating along fibers of the extracellular matrix tend to show a coiling behaviour, i.e. they continuously wrap around the circumference of the fiber while moving forward. This intriguing study by Sadhu and colleagues provides a model that explains this mode of movement. The authors base their model on the presence of membrane-associated proteins that support convex curvature and protrusive forces that are generated by growing actin networks. They present phase diagrams showing the influence of the fiber radius and the density of membrane proteins of a theoretical cell/vesicle attaching and spreading on the fiber. In experiments, dual view light sheet microscopy is used to visualize coiling behaviour of mouse Schwann cells around axons and of C2C12 cells coiling around a fiber. Moreover, predictions of the model are tested for fibers of different diameters as well as cylindrical and elliptical cross-sections.

This is a highly interesting and innovative study that breaks new ground in the analysis of cell locomotion. As a major caveat, I can not really comment on the assumptions and calculations that were used to generate the theoretical model. This should be seen by an expert in modelling.

Major points:

- 1) Fig. 5: Predictions from the theoretical model should be tested also using another cell type that shows different properties (see also point 2: cortex rigidity, adhesion receptor/substrate).

- 2) The discussion is rather short and should be more elaborate. Potential points to discuss could include:
 - i) influence of the coupling strength between membrane proteins and the actin cytoskeleton
 - ii) influence of linear versus branched actin networks in coiling behaviour
 - iii) influence of different rigidities of the actin cortex on coiling
 - iv) influence of different adhesion receptors/substrate pairs on coiling.

Minor point:

- 1) From the text, it is not always clear what kind of fibers (likely fibronectin) and which type of cell has been used for different experiments. Please state this explicitly in all cases.

Reviewer #2 (Remarks to the Author):

I would like to note that the reviewer has expertise in optical imaging systems rather than biophysics models. The editor requested the reviewer to provide an opinion, in particular, to assess the technical aspects of the use of SPIM and diSPIM microscopy. Therefore, an assessment of the biological significance of the model is outside the scope of the reviewer's expertise.

The presented model is solid and technically sound. The authors explained the coiling motion by using a physical model that minimizes the energy associated with binding energy (curvature and attraction forces), active forces, and adhesion. Although the theoretical model is rooted in simple physical principles (energy minimization), the large number of parameters might make the model look not fancy to readers in a broad field. Nevertheless, the authors successfully derive the physical model, conducted rigorous simulations, and identified the phase. Thereby analyzed the (previously unexplained) mechanism as an interplay between active forces, curved membrane proteins, adhesion, etc.

The theoretical model is supported by the experimental observations, which were enabled by the dual-view light-sheet microscopy that provides high spatiotemporal 4D images. Specifically, the isotropic resolution of the diSPIM system provides high-quality cross-sectional images and kymographs consistent with the theoretical model. However, the current version of the manuscript does not include enough details about the experimental systems, and image acquisition/processing.

- The reviewer suggests providing details of dual-view light-sheet microscopy used in the experiment, including information such as resolution, FOV, frame rate, laser wavelength, laser exposure power, etc. The reviewer expects a level of detail that allows readers to reproduce experimental observations reported here.

- Could you improve the image quality for the experimental verification? Seemingly the image quality varies by the sample. For example, the supplementary movie for the Flat Ribbon cross-sectional tracking image looks clear. However, the 3000 nm case possesses high background noise and hard to identify the structure of the cell (Movie S23). The right image of Fig.6C looks segmented. From this image quality, the manual tracking of the protrusive activity has a potential risk of providing a biased result by the person who draws the tracking lines.

- What is the basis of the minimum and maximum bond length used for simulations (lines 127-129)? Do the values used in the simulations such as bending rigidity, and protein-protein attraction strength derived from physical observations? How do these values affect simulation results? Could you explain the appropriateness of the control parameters?

Reviewer #3 (Remarks to the Author):

How cells migrate on fiber geometry substrate is an interesting and timely research question and the underlying mechanism is still poorly understood. In this study, the authors are taking a primarily computational approach with some experimental corroboration to tackle this question. Using advanced imaging such as diSPIM light sheet microscopy and mainly C2C12 myoblast cell line, fiber migration on polystyrene nanofibers of various geometry are observed. Additionally, migration in vivo of dorsal root ganglia neuron during myelination was also observed as another biological example. The main thrust of the study is in the computational model which treat the cell as a vesical of fixed size that can adhere to the fiber of well-defined geometry. Key parameters of the model include cell size, fiber geometry, protrusive force, adhesion receptors and density of membrane curvature promoting proteins. The model appears to be successful in recapitulating different fiber migration behaviour (coiling or finning) and a phase diagram is presented. The experimental data on the angle dynamics on fiber of different geometry are then compared with theoretical model.

Overall, this is an interesting and ambitious study that attempt to address a timely and challenging question. One reservation I have about the manuscript in its current form is that while the modelling aspect appears to be interesting, the experimental corroboration is only limited to different fiber geometry, which is just one of the multiple model parameters. For the revision, I would suggest that systematic experimental perturbations of other model parameters such as adhesion, membrane curvature protein, and cell actomyosin force generation should be performed (methods to perturb these factors are well known in the field). This will help strengthen the validity and applicability of the model to actual biological systems. Another aspect that the revised manuscript can greatly improve is in the visual representation of the 3D imaging data. Figure 6 in particular is quite challenging to interpret and it is difficult to follow what the cells are doing in 3D, or to compare the experimental data with the experiments. I would suggest using surface renderings (e.g. Driscoll...Danuser, Nature Methods 2019) to enhance the visual representation.

Reviewer #4 (Remarks to the Author):

The authors develop a physical model to explain how cell membrane protrusions coil around extracellular fibers. The basic premise of the model is that a population of curved membrane-bound proteins recruit protrusive forces from actin polymerization, and the membrane spreads over an adhesive surface, with a final configuration resulting from minimization of bending energy and adhesion

energies. The authors predict distinct phases of behavior in which the cell protrusion will stretch along the extracellular fiber, or coil around it, depending on the density of membrane-bound proteins and the fiber radius. Their experimental observations qualitatively agree with the predictions of the model and the methodology is sound and well supported. The study makes an important contribution to our understanding of the physical underpinnings of how cells adhere and migrate through three-dimensional environments. Before publication, the following points should be addressed to improve the clarity of the manuscript:

(1) The model simulates a simple vesicle -- not attached to a cell body -- as a simplified representation of a membrane protrusion. The authors should discuss the potential limitations of this simplification, and any qualitative differences expected in the coiling dynamics.

(2) The authors should provide more discussion of the ranges of input parameters used in the simulations and how these compare with reasonable values for the experimental system.

(3) Figure 2: This figure is too small to see many of the relevant details when viewing the document at full size. After “zooming in” to a very high magnification, I was able to see that the blue spheres in the picture contained a grid with color-coded grid points, which were essential to understanding the elements of the model the figure was meant to convey. The figure could be improved by having an inset that shows a subset of the membrane “zoomed in” to show these relevant details.

(4) Figure 3. The authors should describe the criteria by which the phase boundaries (dashed lines) were determined.

(5) Figure 3(d). In the caption, please point the readers back to which equations represent the analytical prediction of the phase boundaries (solid lines).

(6) Figure 5: The panels should be placed in the order in which they are discussed in the text.

(7) Figure 5: Because this figure contains sub-panels with theoretical and experimental data, the caption needs to label very clearly which is which, to avoid confusion.

(8) Figure 6: A more detailed explanation of schematics and how they correspond to kymographs is needed. For example, schematic in Fig. 6c contains a vertical red line, a horizontal red line, and a vertical dashed black line, all of which lack a description in the caption.

Reply to the Referees

Reviewer #1 (Remarks to the Author):

Cells migrating along fibers of the extracellular matrix tend to show a coiling behaviour, i.e. they continuously wrap around the circumference of the fiber while moving forward. This intriguing study by Sadhu and colleagues provides a model that explains this mode of movement. The authors base their model on the presence of membrane-associated proteins that support convex curvature and protrusive forces that are generated by growing actin networks. They present phase diagrams showing the influence of the fiber radius and the density of membrane proteins of a theoretical cell/vesicle attaching and spreading on the fiber. In experiments, dual view light sheet microscopy is used to visualize coiling behaviour of mouse Schwann cells around axons and of C2C12 cells coiling around a fiber. Moreover, predictions of the model are tested for fibers of different diameters as well as cylindrical and elliptical cross-sections.

This is a highly interesting and innovative study that breaks new ground in the analysis of cell locomotion. As a major caveat, I can not really comment on the assumptions and calculations that were used to generate the theoretical model. This should be seen by an expert in modelling.

We thank the referee for the encouraging comments about our manuscript. We reply to each comment of the referee below.

Major points:

1) Fig. 5: Predictions from the theoretical model should be tested also using another cell type that shows different properties (see also point 2: cortex rigidity, adhesion receptor/substrate).

Coiling was observed in other cells types, as was reported in a number of publications, which we already cited in the Introduction section [27,28]. A recent publication from the lab of one of the authors (Nain)

showed the reduced coiling and spreading of cellular protrusions when a curved membrane protein that recruits actin (namely IRsp53) is diminished [48]. This new publication is now added to the Discussion section.

2) The discussion is rather short and should be more elaborate. Potential points to discuss could include:

- i) influence of the coupling strength between membrane proteins and the actin cytoskeleton
- ii) influence of linear versus branched actin networks in coiling behaviour
- iii) influence of different rigidities of the actin cortex on coiling
- iv) influence of different adhesion receptors/substrate pairs on coiling.

These are all excellent points raised by the reviewer, which have now been added to the discussion.

Minor point:

1) From the text, it is not always clear what kind of fibers (likely fibronectin) and which type of cell has been used for different experiments. Please state this explicitly in all cases.

We now mentioned this explicitly in the text and the captions of the figures.

Reviewer #2 (Remarks to the Author):

I would like to note that the reviewer has expertise in optical imaging systems rather than biophysics models. The editor requested the reviewer to provide an opinion, in particular, to assess the technical aspects of the use of SPIM and diSPIM microscopy. Therefore, an assessment of the biological significance of the model is outside the scope of the reviewer's expertise.

The presented model is solid and technically sound. The authors explained the coiling motion by using a physical model that minimizes the energy associated with binding energy (curvature and attraction forces), active forces, and adhesion. Although the theoretical model is rooted in simple physical principles (energy minimization), the large number of parameters might make the model look not fancy to readers in a broad field. Nevertheless, the authors successfully derive the physical model, conducted rigorous simulations, and identified the phase. Thereby analyzed the (previously unexplained) mechanism as an interplay between active forces, curved membrane proteins, adhesion, etc.

The theoretical model is supported by the experimental observations, which were enabled by the dual-view light-sheet microscopy that provides high spatiotemporal 4D images. Specifically, the isotropic resolution of the diSPIM system provides high-quality cross-sectional images and kymographs consistent with the theoretical model. However, the current version of the manuscript does not include enough details about the experimental systems, and image acquisition/processing.

We thank the referee for positive and motivating comments about our work and reply to all the concerns raised by the referee below.

- The reviewer suggests providing details of dual-view light-sheet microscopy used in the experiment, including information such as resolution, FOV, frame rate, laser wavelength, laser exposure power, etc.

The reviewer expects a level of detail that allows readers to reproduce experimental observations reported here.

We now provided all the details about the dual-view light-sheet microscopy used in our experiments, as requested by the reviewer, in section VI.A.3.

- Could you improve the image quality for the experimental verification? Seemingly the image quality varies by the sample. For example, the supplementary movie for the Flat Ribbon cross-sectional tracking image looks clear. However, the 3000 nm case possesses high background noise and hard to identify the structure of the cell (Movie S23). The right image of Fig.6C looks segmented. From this image quality, the manual tracking of the protrusive activity has a potential risk of providing a biased result by the person who draws the tracking lines.

We now improved the image quality of Fig.6 and Movie S23.

- What is the basis of the minimum and maximum bond length used for simulations (lines 127-129)? Do the values used in the simulations such as bending rigidity, and protein-protein attraction strength derived from physical observations? How do these values affect simulation results? Could you explain the appropriateness of the control parameters?

The choice of minimum and maximum bond length is to respect the self-avoidance of the triangulated network. The choice of the parameters is based on our previous study [30], where we mapped the phases of spreading vesicles within this simplified model, and obtained the regime of parameters where protrusions with lamellipodia-like leading edge form. We therefore use here parameters identified in that previous study, so that we are able to simulate the dynamics of such protrusions on the fibers.

We now mention all these points in pages 4-5.

Reviewer #3 (Remarks to the Author):

How cells migrate on fiber geometry substrate is an interesting and timely research question and the underlying mechanism is still poorly understood. In this study, the authors are taking a primarily computational approach with some experimental corroboration to tackle this question. Using advanced imaging such as diSPIM light sheet microscopy and mainly C2C12 myoblast cell line, fiber migration on polystyrene nanofibers of various geometry are observed. Additionally, migration in vivo of dorsal root ganglia neuron during myelination was also observed as another biological example. The main thrust of the study is in the computational model which treat the cell as a vesical of fixed size that can adhere to the fiber of well-defined geometry. Key parameters of the model include cell size, fiber geometry, protrusive force, adhesion receptors and density of membrane curvature promoting proteins. The model appears to be successful in recapitulating different fiber migration behaviour (coiling or finning) and a phase diagram is presented. The experimental data on the angle dynamics on fiber of different geometry are then compared with theoretical model.

We thank the referee for the positive comments about our research work.

Overall, this is an interesting and ambitious study that attempt to address a timely and challenging question. One reservation I have about the manuscript in its current form is that while the modelling aspect appears to be interesting, the experimental corroboration is only limited to different fiber geometry, which is just one of the multiple model parameters. For the revision, I would suggest that systematic experimental perturbations of other model parameters such as adhesion, membrane curvature protein, and cell actomyosin force generation should be performed (methods to perturb these factors are well known in the field). This will help strengthen the validity and applicability of the model to actual biological systems.

In order to address the concerns raised by the referee we varied the adhesion strength of the cell and the fiber, by changing the concentration of fibronectin that is coating the fiber. We present these results in a new SI figure (Fig.S10). From our model, we expected that below a critical adhesion strength, the coiling will cease, as shown in Fig.S10a,b. However, it turns out that cells are able to secrete some minimal level of coating which enables them to spread and coil even on bare plastic fibers. We do find however that the persistence of coiling events was reduced when the adhesion strength decreased. At high levels of adhesion, the ruffles become more strongly adsorbed to the fiber, making it very difficult to follow their coiling motion in the experiment.

These experiments were extremely challenging, both from the preparation aspect, as well as the microscopy and data analysis. While these results were not as conclusive as we hoped, they recover various aspects of the model. We consider them to of interest, thus, we include them in the SI, as they further show the robustness of the coiling phenomenon. We have added these discussions at the end of result section (IV.C.) to reflect these points.

Regarding variation in the curved membrane proteins, we cited our work on IRSp53-KO cells which exhibit reduced coiling (new reference [48]), but did not manage to explore them with the light-sheet microcopy. Similarly, the exploration of the effects of inhibiting myosin-II contractility on the coiling phenomenon will await future studies.

We note that future studies will be needed in order to systematically explore the predictions and validity of our model, as we note at the end of the discussion.

Another aspect that the revised manuscript can greatly improve is in the visual representation of the 3D imaging data. Figure 6 in particular is quite challenging to interpret and it is difficult to follow what the cells are doing in 3D, or to compare the experimental data with the experiments. I would suggest using surface renderings (e.g. Driscoll...Danuser, Nature Methods 2019) to enhance the visual representation.

We followed the suggestions of the referee and improved the quality of Fig.6, while keeping it within the same type of 2D kymograph format which we think is easier to follow.

Reviewer #4 (Remarks to the Author):

The authors develop a physical model to explain how cell membrane protrusions coil around extracellular fibers. The basic premise of the model is that a population of curved membrane-bound

proteins recruit protrusive forces from actin polymerization, and the membrane spreads over an adhesive surface, with a final configuration resulting from minimization of bending energy and adhesion energies. The authors predict distinct phases of behavior in which the cell protrusion will stretch along the extracellular fiber, or coil around it, depending on the density of membrane-bound proteins and the fiber radius. Their experimental observations qualitatively agree with the predictions of the model and the methodology is sound and well supported. The study makes an important contribution to our understanding of the physical underpinnings of how cells adhere and migrate through three-dimensional environments. Before publication, the following points should be addressed to improve the clarity of the manuscript:

We thank the referee for the encouraging comments about our manuscript. We reply to each comment of the referee below.

(1) The model simulates a simple vesicle -- not attached to a cell body -- as a simplified representation of a membrane protrusion. The authors should discuss the potential limitations of this simplification, and any qualitative differences expected in the coiling dynamics.

The reviewer is correct in noting the limitation of our model, which we had mentioned in the original manuscript. As per the suggestion of the reviewer, we have added a detailed discussion of the limitations of our simulation model in the Discussion section.

(2) The authors should provide more discussion of the ranges of input parameters used in the simulations and how these compare with reasonable values for the experimental system.

We now added a discussion about the choice of model parameters used in this work, in the last paragraph of the Model section.

(3) Figure 2: This figure is too small to see many of the relevant details when viewing the document at full size. After “zooming in” to a very high magnification, I was able to see that the blue spheres in the picture contained a grid with color-coded grid points, which were essential to understanding the elements of the model the figure was meant to convey. The figure could be improved by having an inset that shows a subset of the membrane “zoomed in” to show these relevant details.

We thank the referee for the suggestion and provide a zoomed-in section of the vesicle surface in Fig. 2.

(4) Figure 3. The authors should describe the criteria by which the phase boundaries (dashed lines) were determined.

We have now revised substantially the description of Fig.3 and the way we identified the transition lines that are plotted on this figure.

(5) Figure 3(d). In the caption, please point the readers back to which equations represent the analytical prediction of the phase boundaries (solid lines).

We added to the caption that the equations for the analytic approximation of the transition lines are given in the SI section S3.

(6) Figure 5: The panels should be placed in the order in which they are discussed in the text. We now reordered the figures so they appear in the right order, as in the text.

(7) Figure 5: Because this figure contains sub-panels with theoretical and experimental data, the caption needs to label very clearly which is which, to avoid confusion.

We now mentioned explicitly which results are theoretical and which are experimental.

(8) Figure 6: A more detailed explanation of schematics and how they correspond to kymographs is needed. For example, schematic in Fig. 6c contains a vertical red line, a horizontal red line, and a vertical dashed black line, all of which lack a description in the caption

We now explain the cartoons used in Fig.6 in more details in the revised caption.

REVIEWERS' COMMENTS

Reviewer #4 (Remarks to the Author):

The authors have rigorously revised the manuscript in response to reviewer suggestions. The study breaks new ground in the study of cell motility in 3D extracellular matrices.

Reply to referee #4:

Referees comment: The authors have rigorously revised the manuscript in response to reviewer suggestions. The study breaks new ground in the study of cell motility in 3D extracellular matrices.

Our reply: We thank the referee for encouraging comment about our paper.